# Budgeted Bandits for Power Allocation and Trajectory Planning in UAV-NOMA Aided Networks

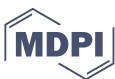

**Ramez Hosny** [1,2,†], **Sherief Hashima** [3,4,*,†] , **Ehab Mahmoud Mohamed** [5,†] , **Rokaia M. Zaki** [1,6,†]
**and Basem M. ElHalawany** [1,7,†]

1    Electrical Engineering Department, Faculty of Engineering at Shoubra, Benha University, Cairo 11614, Egypt;
     r.gad60989@feng.bu.edu.eg (R.H.); rukaia.emam@feng.bu.edu.eg (R.M.Z.);
     basem.mamdoh@feng.bu.edu.eg (B.M.E.)
2    Higher Technology Institute, 10th of Ramadan, Sharkia 44629, Egypt
3    Computational Learning Theory Team, RIKEN-Advanced Intelligence Project, Fukuoka 819-0395, Japan
4    Engineering Department, Nuclear Research Center, Egyptian Atomic Energy Authority, Cairo 13759, Egypt
5    Department of Electrical Engineering, College of Engineering in Wadi Addawasir, Prince Sattam Bin
     Abdulaziz University, Wadi Addawasir 11991, Saudi Arabia; ehab_mahmoud@aswu.edu.eg
6    Higher Institute of Engineering and Technology, Kafr El-Shaikh 33514, Egypt
7    Department of Electronics and Communication Engineering, Kuwait College of Science and Technology,
     Block 4, Doha 13133, Kuwait
*    Correspondence: sherief.hashima@riken.jp
†    These authors contributed equally to this work.

**Abstract:** On one hand combining Unmanned Aerial Vehicles (UAVs) and Non-Orthogonal Multiple Access (NOMA) is a remarkable direction to sustain the exponentially growing traffic requirements of the forthcoming Sixth Generation (6G) networks. In this paper, we investigate effective Power Allocation (PA) and Trajectory Planning Algorithm (TPA) for UAV-aided NOMA systems to assist multiple survivors in a post-disaster scenario, where ground stations are malfunctioned. Here, the UAV maneuvers to collect data from survivors, which are grouped in multiple clusters within the disaster area, to satisfy their traffic demands. On the other hand, while the problem is formulated as Budgeted Multi-Armed Bandits (BMABs) that optimize the UAV trajectory and minimize battery consumption, challenges may arise in real-world scenarios. Herein, the UAV is the bandit player, the disaster area clusters are the bandit arms, the sum rate of each cluster is the payoff, and the UAV energy consumption is the budget. Hence, to tackle these challenges, two Upper Confidence Bound (UCB) BMAB schemes are leveraged to handle this issue, namely BUCB1 and BUCB2. Simulation results confirm the superior performance of the proposed BMAB solution against benchmark solutions for UAV-aided NOMA communication. Notably, the BMAB-NOMA solution exhibits remarkable improvements, achieving 60% enhancement in the total number of assisted survivors, 80% improvement in convergence speed, and a considerable amount of energy saving compared to UAV-OMA.

**Keywords:** UAV; NOMA; trajectory planning; MAB; BUCB; OMA

## 1. Introduction

### 1.1. Background

Rcently, Unmanned Aerial Vehicles (UAV)-enabled wireless communications have witnessed remarkable market growth due to their pros such as low cost, high mobility, ubiquity trajectory, etc. [1,2]. Moreover, their flexibility, huge coverage range, tremendous data rates, and energy consumption can be further extended/improved by optimizing their position trajectory [3,4]. From the communication perspective, leveraging both UAV and Non-Orthogonal Multiple Access (NOMA) technologies requires joint optimization of Power Allocation (PA) and UAV positioning/trajectory [5]. UAV-based emergency communications require an energy-efficient trajectory due to UAVs' limited battery capacity.

Specifically, UAVs can function as an aerial Base Station (BS) to optimize the wireless connectivity of ground nodes by adequately adjusting the UAV location/routes in addition to the transmission parameters such as NOMA.

NOMA is a crucial player in next-generation communication applications including, but not limited to, re-configurable intelligent surfaces [6], millimeter wave and Terahertz communications, power-line communication [7], Internet-of-Things (IOT) [8], and satellite communication [9]. Consequently, in order to deal with the increasing wireless communication traffic, Orthogonal Multiple Access (OMA) algorithms cannot be used, as the transmission bandwidth is limited, requiring customers/survivors to share resources orthogonally. Hence, NOMA can easily tackle the massive bandwidth demand as the survivors can share time frequency and code but orthogonal on each other [10]. Two main NOMA types include Power-Domain NOMA (PD-NOMA) and Code-Domain NOMA (CD-NOMA) where the survivors are allocated different power levels in the former and other codes in the latter. This paper focuses on the PD-NOMA architecture, where multiple signals are multiplexed at the source, as the receivers exploit Successive Interference Cancellation (SIC) to separate different signals.

### 1.2. Paper Motivation

Exploiting NOMA for UAV Trajectory Planning (UTP) networks improves the service offered to ground customers/survivors in emergency communications and disaster zones by serving more survivors with lower latency and better efficiency. However, UTP-PA in UAV-NOMA systems is a critical issue that should be intelligently handled [11]. Different approaches can tackle such a complex problem, including convex/non-convex optimization, heuristic, and Machine Learning (ML) techniques [12,13]. Furthermore, in Software-Defined Networking (SDN)-aided IoT-Fog networks, UTP-PA in UAV-NOMA plays a pivotal role in enhancing network efficiency and performance. UAV trajectory planning optimizes the flight paths of UAVs, enabling efficient data collection, improved network coverage, and dynamic adaptability based on real-time conditions. Power allocation in UAV-NOMA ensures better resource utilization, enhanced throughput, increased connectivity, and improved energy efficiency, allowing multiple IoT devices to share the same time frequency resources simultaneously. Integrating these technologies with SDN's centralized management facilitates intelligent decision-making, fostering seamless communication, reduced latency, and prolonged UAV flight time, ultimately paving the way for more robust and scalable IoT-Fog networks [14,15].

Due to its overwhelming merits and various smart methodologies, ML gained remarkable attention in communication networks [1,16], particularly in online learning techniques such as Multi-Armed Bandits (MABs), which are model-free/stateless Reinforcement Learning (RL) schemes [17]. MABs are excellent candidates to handle trajectory planning optimization issues for UAV-NOMA networks due to their lightweight and online/self-learning capability. This is in contrast to Deep Learning (DL) solutions that require offline training using ground-truth data collected in the environment, whereas bandits are quickly adjustable to environmental variations without any offline training.

MAB is a sequential decision-making methodology where a player, i.e., the UAV in our case, attempts to maximize its cumulative payoff by selecting suitable arms (i.e., survivors/actions) without prior information about any arm. Furthermore, in Budgeted MABs (BMABs), revealing the reward of any arm is associated with paying a cost (i.e., battery consumption in our case). Hence, the player targets to maximize their reward and simultaneously minimize their cost/budget [18–20].

According to the MAB approach, the player selects an action from a set of arms/actions, providing a decision policy that optimizes the expected reward or payoff [21,22]. Please note that the rewards are unknown to the UAV, which has to select the arm/cluster with the highest payoff. MABs can handle this exploration (explore more clusters)-exploitation (sustain the maximum cluster till now) trade-off according to the algorithm policy. Hence, with the MAB assistance, the UAV can decide its trajectory with a maximum sum rate

reward. Due to its lightweight and stateless structure, MABs, especially BMABs are an ideal solution for UTP-PA problems in UAV-NOMA scenarios.

### 1.3. Paper Contribution

This paper proposes an intelligent methodology to optimize the performance of a UAV-enabled NOMA network in a post-disaster setup, where survivors are grouped in multiple clusters. The objective is to design a UTP-PA model that covers all survivors and minimizes battery consumption, which is quite challenging, even for MABs. These challenges include the complex optimization required to jointly optimize user scheduling, power allocation, and resource allocation, adapting to dynamic channel conditions in a mobile UAV environment, balancing exploration and exploitation tradeoffs, handling scalability issues with a large number of users and resources, and managing overhead and latency associated with feedback and decision-making. Efficient MAB-based algorithms need to be developed to tackle these challenges while considering the real-time operation and responsiveness of the system. Inefficient UTP degrades the overall system's performance and incapability to serve all survivors' demands. To handle such a problem, we divided the disaster area into clusters where each cluster contains multiple survivors, and the UAV trajectory should be optimized across these clusters. Our interest in this paper is to optimize UAV positioning/trajectory and power allocation via PD-NOMA utilization. UTP-PA in UAV-NOMA networks is solved using two Budgeted MAB (BMAB) schemes. Specifically, two BMAB versions of Upper Confidence Bound (UCB) are leveraged to handle this issue: the BUCB1 and the BUCB2. The major contributions of this work are outlined as follows:

- The UTP-PA optimization problem of UAV-NOMA systems is formulated as BMABs, where the UAV has to maximize its sum rate by serving more survivors and simultaneously optimizing its battery consumption via efficient power allocation.
- We envision two UCB-aided budgeted algorithms, i.e., BUCB1 and BUCB2, where hovering, flying, and rotational energy consumption are considered.
- Numerical results confirm the superior performance of our envisioned MAB solution for both UAV-NOMA and UAV-OMA scenarios compared with the conventional benchmarks.
- BUCB1-NOMA solution achieved 60% enhancement in the total number of assisted survivors, 80% improvement in convergence speed, and considerable energy consumption compared to UAV-OMA.

### 1.4. Paper Organization

The rest of this paper is structured as follows: Section 2 outlines the related work. Section 3 highlights the studied system model followed by UTP-PA problem formulation. Section 4 discusses the envisioned BUCB1 and BUCB2 algorithms. The numerical results are investigated in Section 5, followed by the concluding remarks in Section 6.

## 2. Related Work

Due to their unique merits and intriguing applications, many UAV-NOMA-related works have been handled recently. Table 1 summarizes the related work and highlights the major contributions of each. In [23], the authors optimized the altitude and PA of NOMA-based UAVs to achieve the maximum achievable sum rate for multi-users via NOMA user-rate gains. However, enhancing spectral and energy efficiency is imperative for achieving the maximum sum rate from UAV-enabled communications. In addition, the deployment of UAVs and power allocation schemes were developed in [24] to improve the performance of a UAV-NOMA network. In order to maximize network sum rates, PA for NOMA is optimized based on the ideal location of the UAV. Moreover [25], UAV trajectory planning and PA optimization could be utilized to operate multiple UAV Base Stations (BSs) at a minimum average rate. This was performed via NOMA without considering UAV movement battery consumption. With the aid of trajectory planning and PA, the

authors of [26] were able to maximize the throughput of a UAV relay system. Nevertheless, they considered fading channel and fixed sensor scenarios, not mobile ones. Furthermore, the authors of [27] optimized the broadcast power allocation, ground customer, and UTP in order to maximize the minimum achievable rate in the downlink NOMA scenario.

**Table 1.** Related work summary.

| Reference | Objective | Contribution |
|---|---|---|
| [23] | Optimize UAV altitude and power | Maximum sum rate for UAV-NOMA users |
| [24] | Improve UAV-NOMA performance | Optimal UAV placement and power allocation |
| [25] | UAV trajectory planning | UAV-NOMA optimal TP and PA |
| [26] | Maximize data rate of UAV relay network | Optimal UAV relay performance |
| [27] | Maximize achievable rate in downlink NOMA Scenario | Optimum power allocation |
| [5] | Interference Mitigation | Uplink UAV-NOMA |
| [6] | NOMA-RIS in RF-UOWC analysis | NOMA-RIS outage performance analysis |
| [7] | Power allocation optimization | NOMA- dual-hop system performance |
| [10] | Optimize subchannel assignment and transit power | UAV-NOMA for uplink IOT |
| [12] | Maximize the sum rate of UAV-NOMA | Optimal UAV trajectory and NOMA precoding |
| [28,29] | Resource allocation and power control for UAV-NOMA | Near-optimal performance using MAB |
| [30] | Optimal trajectory planning in UAV mobile edge computing | Near-optimal trajectory using deep Reinforcement Learning |
| [31] | Optimize UAV trajectory in disaster area | Maximum sum rate using UCB |
| [32] | Optimal UTP and PA | Leverageing MAB to optimize Uplink transmit power |
| [33] | Efficient resource scheduling | Learn effective resource scheduling using new exploration policy of UCB |
| [34] | Apply MABs to optimize UAV energy consumption in disaster area | Maximal number of assisted survivor and prolonged UAV battery |

Recently, UAV-NOMA wireless communication issues have been solved using MABs due to their distinctive benefits. Thus, the authors of [5] mitigated aerial-ground interference in cellular-connected UAV communications via uplink NOMA from the UAV to cellular BSs while sharing the spectrum with existing ground users. The authors of [6] evaluated the performance of NOMA-aided Reconfigurable Intelligent Surfaces (RIS)-assisted hybrid Radio Frequency (RF)- Underwater Optical Wireless Communication (UOWC) system. In one paper [7], the authors investigated NOMA-enhanced dual-hop hybrid communication systems with decode-and-forward relay. Additionally, they proposed a power allocation optimization technique for achieving outage-optimal performance. Furthermore, the authors of [10] applied UAV-NOMA for constructing high-capacity IoT uplink transmission systems to optimize the sub-channel assignment and the uplink transmit power of IoT nodes. Moreover, the authors of [12] proposed a UAV-assisted NOMA network, where the UAV and BS collaborate to serve ground users simultaneously to maximize the sum rate by jointly optimizing the UAV trajectory and NOMA precoding. Also, the authors of [28,29] proposed a MAB solution for the UAV-NOMA system that faces joint resource allocation and power control problems. Using the proposed solution, a distributed resource

allocation and power level can be selected via customers/survivors. The authors of [30] proposed a deep reinforcement learning algorithm for trajectory planning of UAV-aided mobile edge computing. According to [31], the MAB issue, the optimal UAV placement, was determined in order to achieve the network's maximum sum rate. They solved the MAB issue using the UCB scheme. Also, in [32], a MAB-aided solution to find the ideal UTP and PA enhances the network's sum rate with regard to the MAB issue. Furthermore, the authors in [33] propose solutions that follow UCB principles for stochastic MAB. In particular, a new exploration policy was implemented in order to learn resource-efficient scheduling algorithms. As a consequence, the coauthors of this work, in [34] proposed the utilization of MAB schemes to optimize UAV energy consumption in disaster area scenarios. They used the UCB algorithms to solve the UAV optimization problem to find the ideal trajectory without NOMA existence. However, to the best of our knowledge, BMABs have not been exploited in UTP-PA of UAV-NOMA problems despite its practical aspects which motivate this work.

Unlike the pre-mentioned works, we propose BUCB1 / BUCB2 schemes that maximize the data rate and optimize UTP to cover all customers/survivors and minimize battery consumption via efficient power allocation. Both algorithms have the same exploitation behavior, which is the division of the observed rewards over the arms cost. However, their management of exploration is different. BUCB1 assumes prior knowledge of all actions/arms' minimum expected costs (i.e., survivor locations are known prior to estimation), and BUCB2 estimates these costs from previous observations (i.e., unknown survivor locations). Numerical simulations demonstrate the effectiveness of our proposed BMAB solutions in terms of the total number of assisted survivors, energy consumption, and convergence speed compared to benchmark solutions.

## 3. System Model and Problem Formulation

This section discusses the UAV-NOMA system model under consideration. Then UTP-PA problem formulation is further discussed.

### 3.1. UAV-NOMA System Model

In this work, we consider a UAV-based wireless communication system, where NOMA has been exploited for multiple access, as shown in Figure 1. The UAV acts as an aerial base station to assist communications in a disaster area as an emergency network where ground communications infrastructure malfunctioned. To improve the clarity of the analysis, the post-disaster region was evenly divided into $K$ clusters. We investigate the performance of a downlink transmission scenario, where a single UAV base station sweeps a trajectory above the area at an altitude $a$ to serve $K$ clusters of ground survivors as $\mathbb{K} = \{1, 2, \dots, k, \dots, K\}$ defines the set of all existing clusters, and $\mathbb{G}_m = \{k_1, k_2, \dots, k_{t_{m-1}}, k_{t_m}\}$ implies the UAV trajectory, where $k_i \in \mathbb{K}$ is the cluster number inside the region, and $t_m$ is the total number of visited clusters by UAV. The UAV trajectory starts at central cluster $k_1$, assists $t_m$ clusters, and ends at $k_1$ before recharging. Similarly, $G_n[k] = (x_n^k, y_n^k)$ is the ground coordinates of the $n_{th}$ survivors in the $k_{th}$ cluster [29]. The UAV hovers above each cluster to serve multiple survivors, where the corresponding central ground coordinates for any cluster $k_i$ is $W_{k_i} = (x_{k_i}, y_{k_i})$ [35]. The UAV initiates its trip from cluster $k_1$ (charging point), selects the next cluster to serve using BMAB algorithms, and hovers to assist survivors [12], Ref. [36] using NOMA transmission. We assume that $k_{t_m+1} = k_1$, which indicate that the UAV starts and terminates its trajectory at the same cluster for recharging. $\mathcal{G}$ refers to all possible trajectories starting and ending at the central cluster.

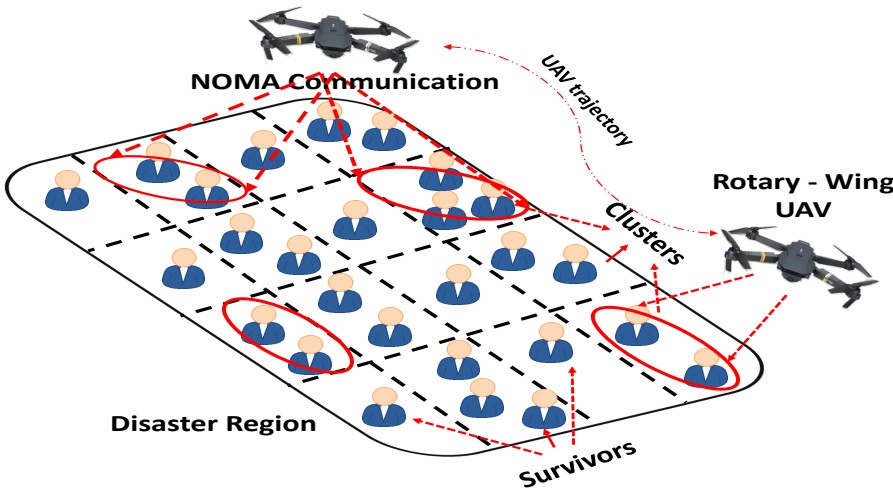

**Figure 1.** A multi-cluster emergency UAV-NOMA enabled network.

The wireless communication channels between UAV and $\mathbb{K}$ clusters are modeled as Rician channels to accommodate for the presence of the Line-Of-Sight (LOS) factor. Therefore, the channel between UAV and the $n_{th}$ survivor in the $k_{th}$ cluster can be modeled as [37]:

$$|h_n[k]|^2 = \frac{\rho_0}{\|\mathbf{G}_n[k] - \mathbf{w}_k\|^2 + a^2} \tag{1}$$

where $\rho_0$ denotes the reference channel power gain with a one-meter reference distance. The total number of stationary survivors is estimated in the area, assuming equal probabilities of seeking radio assistance. $R_{k_{n_{th}}}$ denotes the traffic demands of $n_{th}$ survivors/customers at cluster $k$. A survivor/customer makes an assistance request immediately following a natural disaster. To avoid wasting the UAV energy for assisting a few clusters, the most efficient trajectory that enables serving the survivors' communications while complying with the UAV's battery constraint needs to be optimized. The UAV assists survivors only when it arrives and hovers over a specific cluster, rather than serving them while flying for better channel quality, power, resource allocation, and efficient mobility and trajectory planning. By hovering over the cluster, the UAV can establish better line-of-sight connections, improve channel conditions, allocate power and resources more effectively, and strategically plan its trajectory to conserve energy during flight segments. This approach aims to achieve a balance between communication quality, resource optimization, and energy efficiency within the context of NOMA-based UAV networks. The main objective of the proposed algorithm in this work is to maximize the number of survivors served via a NOMA-based transmission of a single UAV BS, while optimizing the UAV battery consumption/prolonging UAV battery discharging time.

*3.2. UAV-NOMA Transmission Model*

Given a disaster area with multiple clusters, each cluster contains a random number of customers/survivors. Utilizing a UAV exploiting NOMA technology, the task is to jointly optimize its UTP-PA to serve most of the survivors with the least battery consumption. It is possible to treat the issue as an optimization problem, with the aim of maximizing the number of survivors served while decreasing UAV battery consumption. In the downlink transmission of NOMA, the base station transmits signals to multiple survivors simultaneously using the same time and frequency resources. Each survivor is allocated a specific power level and the signals are combined at the receiver side. Through Successive Interference Cancellation (SIC) [6,7], each survivor can decode its own intended signal by sequentially detecting signals with higher received powers and dealing with signals of

users with lower received powers as noise. Therefore, in NOMA, while multiple survivors share the same downlink resources, each survivor's signal can be separated at the receiver.

In the context of a UAV-NOMA communication system, if the channel gains of the survivors of the $k_{th}$ cluster are ordered as $|h_1[k]|^2 > |h_2[k]|^2 > \ldots > |h_n[k]|^2 > \ldots |h_{n_k}[k]|^2$, the received Signal-to-Interference-plus-Noise Ratio (SINR) at the $n_{th}$ survivor, $\forall\, 2 \leq n \leq n_k$, to detect its own message can be mathematically formulated as follows [37]:

$$S_n[k] = \frac{p_{nk}|h_n[k]|^2}{\sum_{i=1}^{n-1} p_{ik}|h_n[k]|^2 + \sigma^2} \tag{2}$$

while the higher-gain user's SINR is given as follows:

$$S_1[k] = \frac{p_{1k}|h_1[k]|^2}{\sigma^2} \tag{3}$$

The power allocated to all the survivors in each cluster can be found sequentially, starting with the higher gain survivor until all power coefficients are found as follows:

$$P_{nk} \geq \delta \left( \sum_{i=1}^{n-1} P_{ik} + \frac{\sigma^2}{|h_n[k]|^2} \right) \tag{4}$$

where $\delta = 2^{\frac{r_{th}}{B}} - 1$ indicates the reliable detection threshold, $r_{th}$ is the rate, $B$ denotes the transmission bandwidth, and $\sigma^2$ represents the noise power at the $n_{th}$ survivor. It is noteworthy that an equal transmit power allocation can also be used to simplify the PA algorithm, which is suitable for many NOMA applications, including IoT, where power control is costly due to the IoT devices' limited capabilities and has been used in much of the literature [28,38]. Consequently, the corresponding achievable sum rate for all assisted survivors in all $K$ clusters can be expressed as:

$$R_K = B \sum_{k=1}^{K} \sum_{n=1}^{n_k} \tau[k] * \log_2(1 + S_n[k]), \tag{5}$$

where $\tau[k]$ is the transmission time for cluster $k$.

### 3.3. UAV's Energy Consumption Model

The UAV consumes energy to perform its tasks, which includes energy for flying from one cluster to the next, hovering over each cluster for a specific period of time, changing direction, and communicating with survivors in each cluster, which can be summarized in the following constraint:

$$t_m P_h T_h + \sum_{t=1}^{t_m} \left( P_f \frac{d_{k_t,k_{t+1}}}{U_f} + \eta_{k_t,k_{t+1}} \right) + \sum_{t=1}^{t_m} \left( P_{max} \tau_t \right) \leq E, \tag{6}$$

where $P_h, U_f, E, T_h, P_f, P_{max}$ are the UAV's hovering power, flying speed, battery capacity, hovering time, average engine flying power, and the maximum allowed power allocated budget to the UAV , respectively. $\tau_t$ denotes the transmit time the UAV allocates to each $k_{th}$ cluster. $d_{k_t,k_{t+1}}$ is the distance between clusters $k_t$ and $k_{t+1}$. $\eta_{k_t,k_{t+1}}$ is the estimated battery consumption of the UAV due to changing its direction to move from cluster $k_t$ to $k_{t+1}$ [39] defined as follows:

$$\eta_{k_t,k_{t+1}} = 2.87 \times 10^{-6}\theta_{k_t}^2 + 4.345 \times 10^{-4}\theta_{k_t} + 0.0026 \quad +0.006\, d_{k_t,k_{t+1}}, \quad (k = 0, 1, \ldots, t_n), \tag{7}$$

where $\theta_{k_t}$ is the angle of the UAV's changing direction given as a function of $\mathbf{p}_{k_i}$ which is the distance between the 2D coordinates of $k_{th}$ cluster. It is mathematically expressed as follows [40]:

$$\theta_{k_i} = \arccos\left( \frac{\langle \overrightarrow{\mathbf{p}_{k_i-1}\mathbf{p}_{k_i}}, \overrightarrow{\mathbf{p}_{k_i}\mathbf{p}_{k_i+1}} \rangle}{\|\overrightarrow{\mathbf{p}_{k_i-1}\mathbf{p}_{k_i}}\|\|\overrightarrow{\mathbf{p}_{k_i}\mathbf{p}_{k_i+1}}\|} \right). \tag{8}$$

### 3.4. Problem Formulation

As the UAV has no prior knowledge of the survivors' data rates and traffic demands, its trajectory should be automatically optimized. In accordance with the traffic demand, UAVs should fly to each cluster and serve the maximum survivors/traffic while underestimating their battery consumption per cluster. This can be performed by observing the survivors' traffic per cluster and battery consumption too.

In the following, we propose a generalized joint optimization problem that optimizes the UAV trajectory $\mathbb{G}_m$, the communication flight time allocation $\{\tau[n]\}$, and the survivor power allocation $\{P_{nk}\}$, which is mathematically formulated as follows:

$$\max_{\{\mathbb{G}_m, P_{nk}, \tau[k]\}} R_K$$

Subject to

$$\sum_{n=1}^{n_k} P_{nk} \leq P_{max}, \forall k \tag{9a}$$

$$0 \leq P_{nk} \leq P_{max}, \forall n, k, \tag{9b}$$

$$t_m P_h T_h + \sum_{t=1}^{t_m} \left( P_f \frac{d_{k_t, k_{t+1}}}{U_f} + \eta_{k_t, k_{t+1}} \right) + \sum_{t=1}^{t_m} \left( P_{max} \tau_t \right) \leq E, \tag{9c}$$

$$t_m \geq K, \tag{9d}$$

where Equations (9a) and (9b) are the constraints for the NOMA power allocation in all clusters, while Equation (9c) is the UAV's battery energy constraint. The restriction Equation (9d) indicates that the UAV visits the whole clusters once at least. We propose an efficient algorithm to find the optimal solution in the following.

## 4. Envisioned BMAB Techniques

The problem of allocating resources (power and time) to different clusters in order to increase the number of customers/survivors served while reducing UAV battery consumption can be formulated as a BMAB problem. In this case, the UAV is the bandit player, the clusters are the bandit arms, and the UAV's power allocation and flight time as the resources to be allocated, i.e., the budget. The reward in this problem is the number of customers/survivors served and the cost is the UAV battery consumption. In BMABs, the goal is to balance the trade-off between exploration and exploitation with cost minimization. Similarly, UAV needs to explore different clusters to gather information about the number of survivors and the battery consumption, while also exploiting the knowledge gained to maximize the reward and minimize the cost [41].

One approach for solving this problem using BMABs is to use the Budget Upper Confidence Bound (BUCB) algorithm. In the BUCB algorithm, exploration and exploitation are balanced by selecting the arm with the highest UCB for the reward, which takes both the average reward and the estimated uncertainty into account [42]. The algorithm also includes a budget constraint to ensure that the UAV's power and flight time do not exceed their maximum limits. Another approach is to use the linearly constrained bandit algorithm, which solves the problem of balancing exploration and exploitation while taking into account the budget constraints. This algorithm uses a linear model to approximate the expected rewards and costs of each arm and solves the problem by solving a linear program in each round. It is worth noting that solving BMAB problems is not a straightforward task,

and it is computationally expensive. Also, It is important to note that these are approximate solutions and the actual results might not be optimal. The success of the solution also depends on the quality of the approximation used and the assumption made about the underlying system.

### 4.1. Proposed UCB Algorithm

UCB is one of the most well-known bandit algorithms for balancing exploration-exploitation compromise [31,42]. The balance between exploration and exploitation is continually updated as it gathers more data about the environment. The first step focuses on exploring all arms, then when the least action trials have occurred, it exploits the arm with the highest calculated payoff. Applying this in UTP-PA problems, the player/UAV selects each arm/cluster once based on the UCB policy. Hence, at every trial $t \in T$, the player draws a arm/cluster $k^* \in \mathbb{K}$ according to the following formula:

$$k_{UCB}^* = \arg\max_{k \in \mathbb{K}} (\overline{R_k}(t) + \sqrt{\frac{2\ln(t)}{\rho_{k,t}}}). \tag{10}$$

where $\overline{R_k}(t)$ refers to the average reward per cluster (i.e., the number of aided customers/survivors) delivered from $k$ cluster at trial $t$, and $\rho_{k,t}$ is the number of times arm/cluster $k$ has been selected. As the cluster is pulled a number of times, the confidence interval enlarges. Hence, the player/UAV attempts other arms/clusters that are less drawn as $\sqrt{2\ln(t)/\rho_{k,t}}$ decreases. As a result of exploiting the past highest-payoff cluster, the player/UAV is able to gain the maximum allowable reward.

### 4.2. Proposed BUCB1/ BUCB2 Algorithms

BMABs classifies into two primary categories: the pure exploration category, referred to as best arm identification, and the exploitation-exploration category [41]. For the first category, only the exploration arms reflect the budget without updating the exploitation arms to determine which arm is best. In contrast to UCB, BUCB1/BUCB2 algorithms represent both exploration and exploitation budgets in the second category.

This allows us to illustrate how the joint UTP and PA problems of the UAV-NOMA system can be solved effectively using BUCB1 and BUCB2 algorithms. In our considered scenario, the UAV cost is a random time variable that should be efficiently anticipated. Furthermore, it is important to reflect both the cost and the payoff of each arm in the exploration-exploitation tradeoff using BUCB1 and BUCB2 algorithms. There is a fundamental difference between the two algorithms in terms of how they manage exploration and explanation [41].

There are two proposed algorithms for BUCB1/BUCB2, both of which have the same exploitation component: the payoff ratio (i.e., the number of customers/survivors) over the costs (i.e., the UAV energy consumption). In BUCB1, the minimum cost of all arms is assumed to be known prior to the game start, so the locations of the survivors are well-known. On the other hand, BUCB2 eliminates this requirement by depending on previous observations to obtain estimated costs. There is a difference between the limits of the proposed algorithms: BUCB2 owns a looser boundary but a wider range than BUCB1 due to the latter requiring more knowledge.

In contrast to the UCB-based UTP-PA algorithm, which has no explicit stopping time, both BUCB1 and BUCB2 cease operation when the energy in the UAV's battery is consumed, as long as the average payoff-to-energy ratio exploitation term remains the same. Hence, the UAV will choose/fly to the cluster with the highest payoffs-to-energy ratio to assist.

Algorithm 1 summarizes the main steps of BUCB1 and BUCB2 schemes. In BUCB1, A parameter $\Delta$ represents the lower bound of expected costs based on prior knowledge [41]:

---

**Algorithm 1:** BUCB1/ BUCB2 Algorithms.

---

    Output:   $k^*_{BUCB1}$,  $k^*_{BUCB2}$
    Input:   $\bar{R}_{k,t}$,  $\bar{C}_{k,t}$,  $\delta$
1  During the first K clusters, each cluster is pulled.
2  # For $z \in u_n$, where $u_n$ is the number of survivors in each $k_{th}$ cluster,
    enforce in Equation (4) the power allocated for the $n_{th}$ survivor on the $k_{th}$
    cluster.
3  The formula below is used to calculate the index $k_{n,t}$ for each $k_{th}$
    arm/cluster.
4  for $t = 1, ...., T$ do
5  # BUCB1
6

$$k^*_{BUCB1,t} = \frac{\bar{R}_{k,t}}{\bar{C}_{k,t}} + \frac{\left(1 + \frac{1}{\Delta}\right)\sqrt{\frac{\ln(t-1)}{\rho_{k,t}}}}{\Delta - \sqrt{\frac{\ln(t-1)}{\rho_{k,t}}}}$$

7  # BUCB2
8

$$k^*_{BUCB2,t} = \frac{\bar{R}_{k,t}}{\bar{C}_{k,t}} + \frac{1}{\Delta_t}\left(1 + \frac{1}{\Delta_t - \sqrt{\frac{\ln(t-1)}{\rho_{k,t}}}}\right)\sqrt{\frac{\ln(t-1)}{\rho_{k,t}}}$$

9  Fly to $k^*_t$, which produces:  $k^*_t = \arg\max_k k^*_{k,t}$.  Then Obtain $R_{k_t}$ and update $C_{k_t}$
    of the selected cluster.
10  End For

---

$$k^*_{BUCB1} = \arg\max_{k \in \mathbb{K}}\left(\frac{\bar{R}_{k,t}}{\bar{C}_{k,t}} + \frac{\left(1 + \frac{1}{\Delta}\right)\sqrt{\frac{\ln(t-1)}{\rho_{k,t}}}}{\Delta - \sqrt{\frac{\ln(t-1)}{\rho_{k,t}}}}\right), \Delta \leq \min_k \mu^C_k, \tag{11}$$

The Global Positioning System (GPS)-based localization makes it simple to obtain this prior knowledge. However, obtaining such knowledge under other scenarios may be difficult if the GPS signal is lost or highly drains the battery in the customer/survivor handset.

Accordingly, BUCB2 analyzes the expected energy of the dispersed survivors/customers on a timely basis, that is, $\Delta_t$, by taking into account the previous energy observations from the clusters that were visited. Thus, BUCB2 utilizes both the minimum necessary cost expectations and the achievable payoffs using empirical observations, as follows [41]:

$$k^*_{BUCB2} = \arg\max_{k \in \mathbb{K}}\left(\frac{\bar{R}_{k,t}}{\bar{C}_{k,t}} + \frac{1}{\Delta_t}\left(1 + \frac{1}{\Delta_t - \sqrt{\frac{\ln(t-1)}{\rho_{k,t}}}}\right)\sqrt{\frac{\ln(t-1)}{\rho_{k,t}}}\right), \Delta_t = \min_k \overline{C}_{k,t}, \tag{12}$$

Following that, the estimate will be used to calculate the exploration term. Thus, this method does not require prior knowledge and can be used in many applications, unlike BUCB1. It is noteworthy that the BUCB2 equation in Algorithm (1) cannot be determined by just substituting $\Delta$ in BUCB1 with $\Delta_t$.

As shown in Algorithm (1), the energy costs are determined by the number of survivors/customers assisted and the input of the next cluster. $\bar{R}_{k,t}$ is the average payoff of cluster k before step $t$, $\bar{C}_{k,t}$ is the average cost, $\rho_{k,t}$ is the time that clusters before step $t$, it has been pulled, and $k_t$ denotes the index of the cluster via algorithm k pulled at time $t$. When a cluster is pulled many times, the confidence interval expands, causing $\sqrt{\ln(t-1)/\rho_{k,t}}$ to decrease and the player/UAV to try other less drawn arms/clusters. The player utilizes the previous highest-payoff cluster to gain the maximum allowable payoff.

## 5. Numerical Simulations

Herein, we evaluate the performance of the proposed two algorithms (BUCB1, BUCB2) based on the UAV-NOMA network, assuming an equal power allocation within each cluster. Then, we will compare their performance with respect to a conventional UAV-OMA scenario. The survivors are deployed randomly within each cluster, assuming their traffics follow Binomial distributions $B\ (u_k, o)$, where $u_n$ is the number of survivors in the $k_{th}$ cluster and $o$ is the on-demand radio access probability that equals to 0.2 (Table 2).

**Table 2.** Simulation parameters.

| Parameter | Value |
|:---:|:---:|
| $U_f$ | 20 Km/h |
| $T_h$ | 120 s |
| $a$ | 10 m |
| $p_h$ | 4 |
| $p_f$ | 2 |
| $k \times k$ | $100 \times 100$ m$^2$ |
| $P_t$ | 40 dBm |
| $B$ | 100 MHZ |
| $\sigma^2$ | $-174$ dBm/HZ |
| $\rho_0$ | $-50$ dB |
| Aerial coverage range | 100 m |
| Channel type | Rician |

In this simulation, the following parameters are used; The UAV's altitude is fixed at a = 10 m, where a default bandwidth of 100 MHz is used for transmission. A maximum power of $P_t$ = 40 dBm is used by the the UAV. Herein, we utilized X-NOMA corresponds to a NOMA scheme with equal power allocation for all survivors in each cluster, while X-NOMA PA denotes a NOMA scheme that uses the power allocation strategy in Equation (4) such that $X \in \{UCB, BUCB1, BUCB2\}$.

Figure 2 shows the number of assisted survivors of the three compared algorithms (i.e., UCB, BUCB1, and BUCB2) for both UAV-NOMA and UAV-OMA with E = 1000 Joule (J) against the convergence time horizon. Comparing the convergence provides valuable insights for algorithm selection. The results highlight the benefit of exploiting NOMA transmission compared with OMA, where all three algorithms achieve much higher speed through using NOMA. On the other hand, the results show that BUCB1 performs best, owing to its precise selection policy with GPS survivors' locations. BUCB2 achieves less performance since it has no access to the prior knowledge of BUCB1. As shown in Table 3, at $t$ = 200 BUCB1, BUCB2, and UCB for UAV-NOMA achieves higher number of aided survivors by 109%, 133%, and 260% compared to similar schemes in UAV-OMA, respectively.

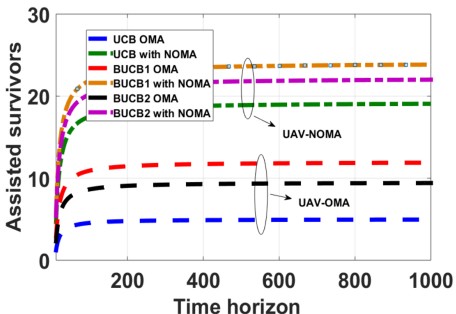

**Figure 2.** Number of assisted survivors versus time horizon.

**Table 3.** Number assisted survivors at $t = 200$.

| | UCB | 5 |
|---|---|---|
| UAV-OMA | BUCB1 | 11 |
| | BUCB2 | 9 |
| | UCB | 18 |
| UAV-NOMA | BUCB1 | 23 |
| | BUCB2 | 21 |

Figure 3 shows the number of assisted survivors versus various UAV transmitted power levels ranging from 10 to 40 dBm at E = 1000 J. For all compared schemes as the power increases the number of assisted survivors gradually increases, especially the UAV-NOMA-related schemes. The BUCB1-NOMA algorithm assists the highest possible number of survivors. As shown in Table 4, at $P_t = 40$ dBm, BUCB1-NOMA. BUCB2-NOMA, and UCB-NOMA performance is better than similar techniques using OMA by 92%, 74%, and 70%, respectively.

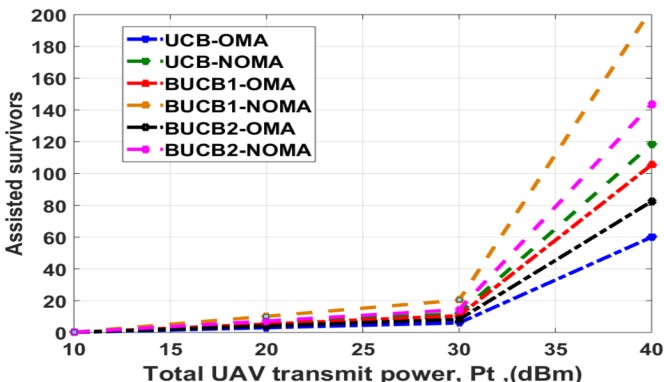

**Figure 3.** Number of assisted survivors versus transmit Power $P_t$, dBm.

**Table 4.** UAV transmit power at $P_t = 40$ dBm.

| | UCB | 69.7 |
|---|---|---|
| UAV-OMA | BUCB1 | 105.5 |
| | BUCB2 | 93.46 |
| | UCB | 118.5 |
| UAV-NOMA | BUCB1 | 203.6 |
| | BUCB2 | 163.3 |

Figure 4 previews the number of assisted survivors versus the number of visited clusters at $E = 1000$ J. The whole compared schemes increase relatively with the number of visited clusters. With a longer flight duration throughout minimizing battery consumption, UAV transmits more information to the clusters. Specifically, the envisioned BUCB1 owns the best performance, followed by BUCB2 and then UCB. Moreover, UAV-NOMA improves performance better than UAV-OMA. The overall number of survivors increases slightly with an increase in the number of clusters, especially in BUCB1 and BUCB2. A more significant number of visited clusters leads to more flying power consumption via the UAV. As a result, the algorithms we offer have an effective energy management strategy. At $E = 1000$ J the number of assisted survivors for the NOMA scenario compared schemes is larger than what is in another case because of the larger battery capacity, the hovering and flying times are longer in the area. BUCB1 performs best, followed by BUCB2 due to its appropriate techniques for both battery capacity scenarios. BUCB1 owns the exact locations of the survivors via GPS. As shown in Table 5, at 25 clusters the UAV-NOMA-BMAB schemes outperform UAV-OMA-BMAB by 27.47%, 24.02%, and 18.29% for BUCB1, BUCB2, and UCB, respectively.

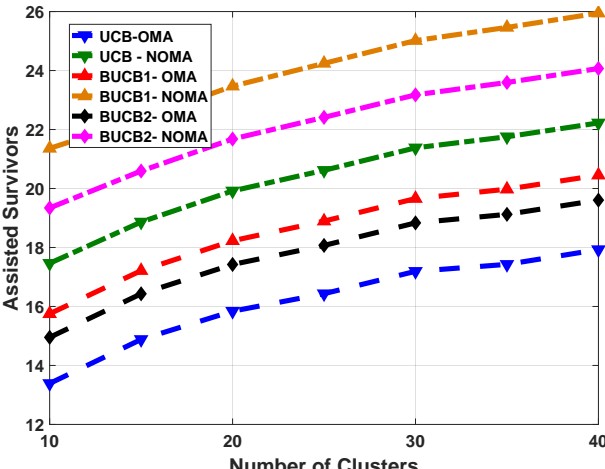

**Figure 4.** Number of assisted survivors versus a number of clusters $\mathbb{N}$ for E = 1000 J.

**Table 5.** Number of assisted survivors for the compared schemes at 25 clusters.

|          |       |       |
|----------|-------|-------|
| UAV-OMA  | UCB   | 17.43 |
|          | BUCB1 | 19    |
|          | BUCB2 | 18.07 |
| UAV-NOMA | UCB   | 20.62 |
|          | BUCB1 | 24.24 |
|          | BUCB2 | 22.41 |

Figure 5 presents the number of survivors/users versus the number of clusters for different NOMA power allocation schemes. The power allocation strategy in Equation (4) denoted by X-NOMA PA show a better performance compared with their counterparts using the equal power allocation schemes denoted by X-NOMA. This help in focusing on exploring better channels. Table 6 reveals the percentage improvements for UCB, BUCB1, and BUCB2, when using the PA strategy in Equation (4), which are approximately 333%, 80%, and 153%, respectively, as the number of survivors at $k = 25$ in UAV-NOMA compared to UAV-OMA, respectively.

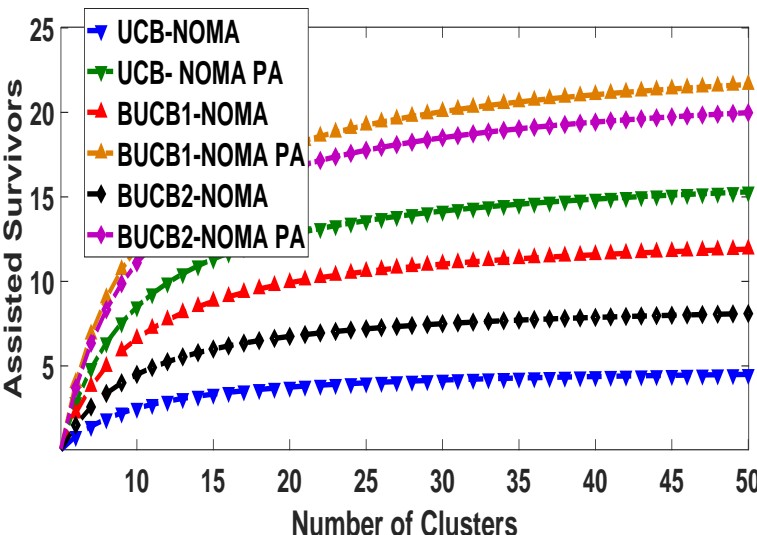

**Figure 5.** Number of assisted survivors versus the number of clusters for different NOMA power allocation schemes.

**Table 6.** Effect of the NOMA PA on the number of survivors at 25 clusters.

|  |  |  |
|---|---|---|
|  | UCB | 3 |
| UAV-NOMA | BUCB1 | 10 |
|  | BUCB2 | 6.5 |
|  | UCB | 13 |
| UAV-NOMA PA | BUCB1 | 18 |
|  | BUCB2 | 16.5 |

Figure 6 shows the effect of using the NOMA power allocation strategy in Equation (4) compared the equal power allocation of NOMA strategy on the assisted survivors' performance of UCB, BUCB1, and BUCB2 algorithms at E = 1000 J over time. The results show that implementing the NOMA power allocation improves the performance significantly compared to the equal power allocation for all schemes. The NOMA PA is expected to outperform equal PA in UAV-NOMA scenarios, where leveraging adaptive PA for resource efficiency with favorable channel conditions, leading to increasing assisted survivors over time. As shown in Table 7 BMAB-NOMA PA techniques outperform similar BMAB-NOMA techniques with eqial PA by approximately 48.77%, 62.73%, and 50.49%, for UCB1, BUCB1, and BUCB2, respectively.

**Table 7.** Power allocation effects on the number assisted survivors at t = 100.

|  |  |  |
|---|---|---|
|  | UCB | 17.53 |
| UAV-NOMA | BUCB1 | 22.3 |
|  | BUCB2 | 20.22 |
|  | UCB | 26.1 |
| UAV-NOMA PA | BUCB1 | 36.25 |
|  | BUCB2 | 30.45 |

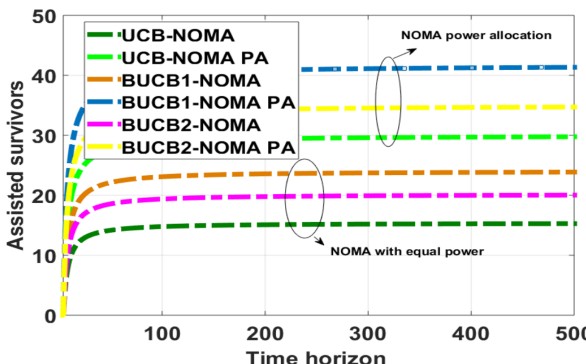

**Figure 6.** Number of assisted survivors versus time horizon NOMA power allocation.

Figure 7 compares UCB, BUCB1, and BUCB2 algorithms for UAV-NOMA under two power allocation scenarios: equal power allocation and NOMA power allocation. Equal power allocation provides fixed power to each survivor in the cluster, while NOMA power allocation allocates varying power levels based on channel conditions and QoS requirements, exploiting multi-survivor diversity. With increasing UAV transmitted power, the assisted survivors may increase linearly, but limitations can arise due to varying channel conditions. The performance of algorithms under NOMA PA depends on channel conditions and power allocation policies and influences the number of visited clusters.

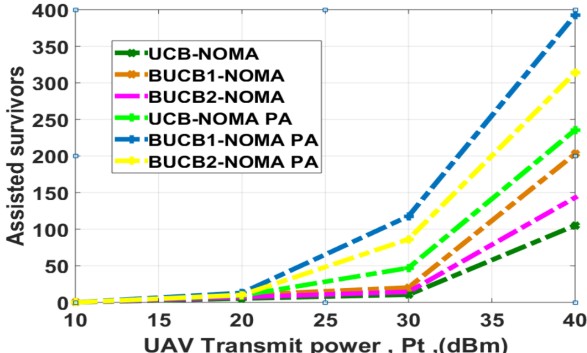

**Figure 7.** Number of assisted survivors versus UAV-NOMA transmit Power $P_t$, dBm.

Table 8 illustrates distinct variations in assisted survivors as UAV transmission power increases. at $P_t = 40$ dbm, BUCB1 with NOMA PA achieves 92.15%, followed by BUCB2 with NOMA PA at 85.28%, and UCB with NOMA PA trailing at 73.32%. Finally, BUCB1-NOMA PA aims to achieve a more balanced allocation of power among survivors in the cluster compared to UCB, and BUCB2, ensuring a fair distribution of resources while maximizing the overall sum rate.

**Table 8.** Effect of the UAV transmit power on the number of assisted survivors at $p_t = 40$ dBm.

|  | UCB | 118.55 |
| --- | --- | --- |
| UAV-NOMA | BUCB1 | 203.6 |
|  | BUCB2 | 163.37 |
|  | UCB | 205.6 |
| UAV-NOMA PA | BUCB1 | 392 |
|  | BUCB2 | 304 |

## 6. Conclusions

In this paper, we proposed a novel BMAB-based approach for power allocation and trajectory planning in UAV-NOMA-aided networks that has shown promising results in optimizing the performance of such networks. Using BMAB algorithms, UAVs can efficiently allocate power and determine optimal trajectories based on available information and environmental feedback. BMABs balance exploration and exploitation and consider the UAV battery budget leading to better decision-making and improved communication performance. Hence, we proposed two UCB-aided budgeted algorithms, i.e., BUCB1 and BUCB2, that effectively assisted more survivors in disaster area scenarios. Hence, UAV-NOMA networks can enhance spectral efficiency and support multiple clusters simultaneously. The proposed algorithms were utilized to allocate power and optimize UAV positioning/trajectory, further improving UAV network performance and making them more efficient and effective in various applications. BMAB-NOMA achieved remarkable improvements, with a 60% increment in assisted survivors, 80% enhancement in convergence, and significant energy saving compared to the UAV-OMA solution. These findings underscore the BMAB approach's effectiveness, efficiency, and potential to significantly elevate UAV-NOMA power allocation performance. Future directions might include inspecting UAV-NOMA in SDN-IOT Fog networks with multiplayer UAV scenario.

**Author Contributions:** Conceptualization, S.H. and B.M.E.; Formal analysis, S.H.; Investigation, R.H., S.H., E.M.M., R.M.Z. and B.M.E.; Supervision, S.H., E.M.M., R.M.Z. and B.M.E.; Validation, S.H., E.M.M. and R.M.Z.; Writing—original draft, R.H., S.H. and B.M.E.; Writing—review & editing, R.H., S.H., E.M.M. and B.M.E. All authors have read and agreed to the published version of the manuscript.

**Funding:** This work was supported by JSPS KAKENHI Grant Numbers JP21K14162 and JP22H03649 Japan. It is also supported via funding from Prince Sattam bin Abdulaziz University project number (PSAU/2023/R/1444) KSA.

**Data Availability Statement:** Data are available upon request to the authors.

**Conflicts of Interest:** The authors declare no conflict of interest.

## Abbreviations

The following abbreviations are used in this manuscript:

| Acronym | Abbreviation |
| --- | --- |
| UAV | Unmanned Aerial Vehicle |
| NOMA | Non Orthogonal Multiple Access |
| OMA | Orthogonal Multiple Access |
| PA | Power Allocation |
| TP | Trajectory Planning |
| BMAB | Budget Multi-Armed Bandit |
| PD-NOMA | Power domain NOMA |
| CD-NOMA | Code domain NOMA |
| SDN | Software-defined networking |
| UCB | Upper Confidence Bound |
| BS | Base Station |
| RF | Radio Frequency |
| UOWC | Underwater Optical Wireless Communication |
| LOS | Line Of Side |
| ML | Machine Learning |
| RL | Reinforcement Leaning |
| DL | Deep Learning |
| SIC | Successive Interference Cacellation |
| QoS | Quality of Service |
| SINR | Signal-to-Interference-Plus Noise Ratio |
| GPS | Global Positioning System |

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
