# Peer review of "Budgeted Bandits for Power Allocation and Trajectory Planning in UAV-NOMA Aided Networks"

_drones, doi:10.3390/drones7080518_

Round 1
Reviewer 1 Report
To help many survivors in a post-disaster scenario where ground stations are malfunctioning, the authors research efficient Power Allocation and Trajectory Planning Algorithms for UAV-aided NOMA systems.
1- Please write the name of the tool you used for evaluation and the percentage of improvements in the abstract, introduction, and conclusion.
2- Please divide the introduction into some subsections: Motivation, Contribution, and Organization.
3- I recommend the authors add reference architecture to the article.
4- What is the communication type you used? Did you use D2D or a broker approach?
5- How can we use your approach in SDN-based IoT-Fog networks?
6- How can we use your approach in SDN-based UAV-NOMA Aided Networks? You can cite some articles about "SDN-based IoT-Fog networks" and explain your approach to possible SDN-based UAV-NOMA Aided Networks.
Fingers crossed.
Author Response
Reviewer#1: To help many survivors in a post-disaster scenario where ground stations are malfunctioning, the authors research efficient Power Allocation and Trajectory Planning Algorithms for UAV-aided NOMA systems
Author response: We would like to deeply thank the respected reviewer for the time and efforts she/he spent in reviewing our paper. Also, thanks so much for finding the research topic presented in our humble paper interesting and the document is well structured.
Author action: We tried our best to cope with all comments/concerns given by the respected reviewer as follows.
Reviewer#1, Concern # 1: 1) Please write the name of the tool you used for evaluation and the percentage of improvements in the abstract, introduction, and conclusion.
Author response: We would like to deeply thank the respected reviewer for this valuable and constructive comment, we have reflected the improvement percentage. We added the percentage improvement as advised by repected reviewer
Author action: We have fixed it in the revised manuscript as follows:
Revised Manuscript
”
Reviewer#1, Concern # 2 Please divide the introduction into some subsections: Motivation, Contribution, and Organization.
Author response: We would like to deeply thank the respected reviewer for this valuable and constructive comment. We divided the introduction to subsection for ease redability by readers.
Reviewer#1, Concern # 3: I recommend the authors add reference architecture to the article.
Author response: We would like to deeply thank the respected reviewer for this valuable and constructive comment. We added related work summary table in section 2.
Author action: To cope with this valuable comment, we have added the following table in the revised manuscript:
“
Revised Manuscript
”
Reviewer#1, Concern # 4: What is the communication type you used? Did you use D2D or a broker approach?
Author response: We would like to deeply thank the respected reviewer for this valuable comment. Here we utilized UAV in disaster area scenario to assist multiple survivors using NOMA technique to prolong the battery utilization and assist more survivors. We did not used D2D or broker schemes in this work.
Reviewer#1, Concern # 5: How can we use your approach in SDN-based IoT-Fog networks?
Author response: We would like to deeply thank the respected reviewer for this valuable comment.
In SDN-based IoT-Fog networks, UAV trajectory planning and power allocation in UAV-NOMA offer significant improvements in efficiency and performance. By optimizing UAV trajectories, data collection and delivery become more efficient, ensuring better network coverage and reduced energy consumption. Power allocation in UAV-NOMA enhances resource utilization, throughput, and connectivity, benefiting from simultaneous data transmission and increased spectral efficiency. The integration of trajectory planning with power allocation, facilitated by SDN's centralized control, enables dynamic adaptation and real-time decision-making, making UAVs an indispensable component for data collection, communication, and network optimization in future IoT-Fog ecosystems.
Author action: To cope with this valuable comment, we have added the following table in the revised manuscript:
“
Revised Manuscript
Reviewer#1, Concern # 6: How can we use your approach in SDN-based UAV-NOMA Aided Networks? You can cite some articles about "SDN-based IoT-Fog networks" and explain your approach to possible SDN-based UAV-NOMA Aided Networks.
Author response: We would like to deeply thank the respected reviewer for this valuable comment. We added this as future work for deep investigations.
Author action: To cope with this valuable comment, we have added the following sentences in the revised manuscript:
“
Revised Manuscript
”

Reviewer 2 Report
The topic is interesting and important. However, the problem formulation can be think over again and the solutions and simulations can be further improved. Detailed comments are given as follows:
1. It seems that the explanation for CD-NOMA and that for PD-NOMA are reversed in line 35, page 2.
2. Why UTP-PA with NOMA is critical? The authors should clearly illustrate the importance, difference, and difficulty of the UTP-PA with NOMA compared with other benchmark methods, like UTP-PA without NOMA, UTP with NOMA, UAV with NOMA, etc...
3. What is the main challenges of the MAB based design when NOMA is considered in the UTP-PA problem in the UAV networks?
4. The symbols in formula (1) are not properly explained. Other similar problems, e.g., r_{th} in (4), t_{m} in (6).
5. Many super/sub-scripts are confusing, like the k_{th} and k^{th}, n_{th} and n_{t}, \tau_{t}and \tau[n]...
6. Since downlink NOMA is considered in this paper, the denominator of formula (2) is not correct. h_{i}[k] should be changed to h_{n}[k].
7. What is the relationship between the communication flight time \tau_{t} and the hovering time T_[h]?
8. According to the illustration of the authors, the UAV serves the users via NOMA in a certain cluster only when it has arrived and hovered over this cluster. The reviewer wonders why the UAV does not serve the users while it is flying?
9. The UAV trajectory should satisfy certain constraint since the locations of the UAV in different time slots are sequential. It seems that this aspect is not considered in the formulated problem (9).
10. In the numerical simulation part, equal power allocation within each cluster is assumed, which is not consistent with the original problem formulation where both trajectory, PA, and the serving time for each cluster is jointly studied. The authors should add more corresponding simulations and analysis.
The quality of language is satisfied.
Author Response
Reviewer#2: The topic is interesting and important. However, the problem formulation can be think over again and the solutions and simulations can be further improved. Detailed comments are given as follows:
Author response: We would like to deeply thank the respected reviewer for the time and efforts she/he spent in reviewing our paper. Also, thanks so much for finding the research topic presented in our humble paper interesting and the document is well structured.
Author action: We tried our best to cope with all comments/concerns given by the respected reviewer as follows.
Reviewer#2, Concern # 1: It seems that the explanation for CD-NOMA and that for PD-NOMA are reversed in line 35, page 2.
Author response: We would like to deeply thank the respected reviewer for this deep focus. We corrected this point in the revised manuscript
Author action: To cope with this valuable comment, we have added the following sentences in the revised manuscript:
“
Revised Manuscript
”
Reviewer#2, Concern # 2: Why UTP-PA with NOMA is critical? The authors should clearly illustrate the importance, difference, and difficulty of the UTP-PA with NOMA compared with other benchmark methods, like UTP-PA without NOMA, UTP with NOMA, UAV with NOMA, etc...
Author response: We would like to deeply thank the respected reviewer for this valuable comment. We improved the paper motivation and showed why this problem is critical
Author action: To cope with this valuable comment, we have added the following table in the literature review section of the revised manuscript.
“
Revised Manuscript
Reviewer#2, Concern # 3: What is the main challenges of the MAB based design when NOMA is considered in the UTP-PA problem in the UAV networks?
Author response: We would like to deeply thank the respected reviewer for this valuable comment. Incorporating NOMA in the context of UAV trajectory planning and power allocation (UTP-PA) problems in UAV networks poses several challenges for Multi-Armed Bandit (MAB) based designs. These challenges include the complex optimization required to jointly optimize user scheduling, power allocation, and resource allocation, adapting to dynamic channel conditions in a mobile UAV environment, balancing exploration and exploitation tradeoffs, handling scalability issues with a large number of users and resources, and managing overhead and latency associated with feedback and decision-making. Efficient MAB-based algorithms need to be developed to tackle these challenges while considering the real-time operation and responsiveness of the system.
Author action: To cope with this valuable comment, manuscript as follows.
Reviewer#2, Concern # 4: The symbols in formula (1) are not properly explained. Other similar problems, e.g., r_{th} in (4), t_{m} in (6).
Author response: We would like to deeply thank the respected reviewer for this valuable comment. We explained them in the revised manuscript.
Author action: To cope with this valuable comment, we have added the following in the revised manuscript:
“
Revised Manuscript
”
Reviewer#2, Concern # 5: Many super/sub-scripts are confusing, like the k_{th} and k^{th}, n_{th} and n_{t}, \tau_{t}and \tau[n]...
Author response: We would like to deeply thank the respected reviewer for this valuable comment. We precisely revised the manuscript and unified these notations
Reviewer#2, Concern # 6: Since downlink NOMA is considered in this paper, the denominator of formula (2) is not correct. h_{i}[k] should be changed to h_{n}[k]
Author response: We would like to deeply thank the respected reviewer for this valuable and constructive comment. The equation is corrected in the revised manuscript.
Author action: To cope with this valuable comment, we have updated formula 2 in the revised manuscript as follows:
“
Revised Manuscript
”
Reviewer#2, Concern # 7: What is the relationship between the communication flight time \tau_{t} and the hovering time T_[h]?
Author response: We would like to deeply thank the respected reviewer for this valuable and constructive comment. The relationship depends on the specific application, mission requirements, and operational constraints. Generally, the communication flight time refers to the duration during which the UAV is actively engaged in data transmission or reception, while the hovering time refers to the duration the UAV remains stationary at a specific location. In some scenarios, the communication flight time and hovering time can be inversely related. For example, if the UAV needs to cover a large area and maintain communication with multiple ground users or base stations, it may have shorter hovering time as it needs to allocate more time for flying and maintaining connectivity. On the other hand, if the UAV's primary objective is to provide prolonged communication services to a specific location or a group of users, it may have longer hovering time and shorter communication flight time. However, it's important to note that this relationship is highly dependent on the specific mission requirements and constraints. The UAV's trajectory planning, power allocation, battery capacity, data transmission rates, and operational limitations all influence the distribution of communication flight time and hovering time. Therefore, the relationship between \tau_{t} and T_[h] is not fixed and must be carefully optimized based on the objectives and constraints of the UAV network application.
Reviewer#2, Concern # 8: According to the illustration of the authors, the UAV serves the users via NOMA in a certain cluster only when it has arrived and hovered over this cluster. The reviewer wonders why the UAV does not serve the users while it is flying?
During flying no robust communication links after disaster the users/survivors are randomly distributed. (high dynamic channel conditions coming from UAV flight disturbance and it may drain the UAV and users batteries without sustaining a robust communication links)
Author response: We would like to deeply thank the respected reviewer for this valuable and constructive comment. The authors' decision for the UAV to serve users only when it has arrived and hovered over a specific cluster, rather than serving them while flying, can be attributed to considerations such as optimizing channel quality, power and resource allocation, and efficient mobility and trajectory planning. By hovering over the cluster, the UAV can establish better line-of-sight connections and improve channel conditions, allocate power and resources more effectively, and strategically plan its trajectory to conserve energy during flight segments. This approach aims to achieve a balance between communication quality, resource optimization, and energy efficiency within the context of NOMA-based UAV networks.
Author action: To cope with this valuable comment, we have added this scentence in the revised manuscript as follows:
“
Revised Manuscript
Reviewer#2, Concern # 9: The UAV trajectory should satisfy certain constraint since the locations of the UAV in different time slots are sequential. It seems that this aspect is not considered in the formulated problem (9).
Author response: We would like to deeply thank the respected reviewer for this valuable and constructive comment. A user usually needs to be served only once in the real post-earthquake scenario so serving a few grids repeatedly would cause power waste. Then we can assign a weight factor to the traffic of the visited grids before to reduce the probability of visiting the grids again.
Author action: To cope with this valuable comment, we have updated the optimization problem with new constraint:
Revised Manuscript
”
Reviewer#2, Concern # 10: In the numerical simulation part, equal power allocation within each cluster is assumed, which is not consistent with the original problem formulation where both trajectory, PA, and the serving time for each cluster is jointly studied. The authors should add more corresponding simulations and analysis
Author response: We would like to deeply thank the respected reviewer for this valuable and constructive comment. We updated the numerical simulations

Reviewer 3 Report
The article titled “Budgeted Bandits for Power Allocation and Trajectory Planning in UAV-NOMA Aided Networks”, relates to my area of interest that’s why I suggest or recommend some points which may help in order to improve the readability as well as overall structure of this manuscript. The following are my suggestions, recommendations and questions for this article which may help to improve the quality of this manuscript are as follows.
1. In title of this article need to update regarding your work done.
2. Abstract
· Must use the contrast word i.e., On one hand and On the other hand.
· By using the comparison of the indicators, how much percent the algorithm is efficient.
3. Introduction
· Use the citations in the end of paragraph is best.
· What is the main motivation of this research elaborate it?
· Line 43 please check the number of citations used
· Figure 1 made it yourself if it is no need to write the citation in the figure. It’s best to write in the paragraph where figure detail is written.
· Line 52-56 has some flow of references.
· Introduction must split into subheadings (Background, Motivation, Contribution and Organization).
4. Related Work
· Related work is too short, please add some more work.
· Give some pictorial representation i.e figures.
· Create a short table for related work in order to understand your contributions.
5. System Model and Problem Formulation and ENVISIONED BMAB TECHNIQUES
· In this section explanation of the system model seems difficult for general user oplease make it user friendly.
· Transmission models should have figures to explain.
· Please check the format (font) in section 4
6. Results
· The quality of figures 2 and 3 is not as per the standard.
· Line 310-314 it is recommended to make a proper comparison table or figure.
· Tables are missing in the result section.
7. General Comments
· After the references insert List of abbreviations.
· In the introduction first para very, less citations are there Need to revise it properly.
· Data Availability Statement: Not applicable.
· Check all the references carefully.
Overall English is better in this article. just need to recheck some minor spell checking and grammatical errors.
Author Response
Reviewer#1: To help many survivors in a post-disaster scenario where ground stations are malfunctioning, the authors research efficient Power Allocation and Trajectory Planning Algorithms for UAV-aided NOMA systems
Author response: We would like to deeply thank the respected reviewer for the time and efforts she/he spent in reviewing our paper. Also, thanks so much for finding the research topic presented in our humble paper interesting and the document is well structured.
Author action: We tried our best to cope with all comments/concerns given by the respected reviewer as follows.
Reviewer#3, Concern # 1: In title of this article need to update regarding your work done.
Author response: We would like to deeply thank the respected reviewer for this valuable and constructive comment. The current title is almost close to what we have done otherwise the respected reviewer should provide us with his suggested title.
Reviewer#3, Concern # 2: Abstract
- Must use the contrast word i.e., On one hand and On the other hand.
- By using the comparison of the indicators, how much percent the algorithm is efficient.
Author response: We would like to deeply thank the respected reviewer for this valuable and constructive comment. The abstract is updated
Author action: To cope with this valuable comment, we have updated the abstract in the revised manuscript as follows:
“Revised Manuscript
Reviewer#3, Concern # 3: Introduction
- Use the citations in the end of paragraph is best.
- What is the main motivation of this research elaborate it?
- Line 43 please check the number of citations used
- Figure 1 made it yourself if it is no need to write the citation in the figure. It’s best to write in the paragraph where figure detail is written.
- Line 52-56 has some flow of references.
- Introduction must split into subheadings (Background, Motivation, Contribution and Organization).
Author response: We would like to deeply thank the respected reviewer for this valuable and constructive comment. We carefully checked the manuscript and reflected those comments on the revised version
Author action: To cope with this valuable comment, we have updated the conclusion section in the revised manuscript as follows:
“
Revised Manuscript
”
Reviewer#3, Concern # 4: Related Work
- Related work is too short, please add some more work.
- Give some pictorial representation i.e figures.
- Create a short table for related work in order to understand your contributions.
Author response: We would like to deeply thank the respected reviewer for this valuable and constructive comment. The conclusion section is updated in the revised manuscript as recommended by the respected reviewer.
Author action: To cope with this valuable comment, we have updated the related work section in the revised manuscript as follows:
“
Revised Manuscript
”
Reviewer#3, Concern # 5: System Model and Problem Formulation and ENVISIONED BMAB TECHNIQUES
- In this section explanation of the system model seems difficult for general user oplease make it user friendly.
- Transmission models should have figures to explain.
- Please check the format (font) in section 4
Author response: We would like to deeply thank the respected reviewer for this valuable and constructive comment. We revised the manuscript and corrected all of these flows
”
Reviewer#3, Concern # 6: Results
- The quality of figures 2 and 3 is not as per the standard.
- Line 310-314 it is recommended to make a proper comparison table or figure.
- Tables are missing in the result section.
Author response: We would like to deeply thank the respected reviewer for this valuable and constructive comment. We improved the quality of the figures and added performance tables for comparison in the results section
Author action: To cope with this valuable comment, we have updated the conclusion section in the revised manuscript as follows:
“
Revised Manuscript
”
Reviewer#3, Concern # 7: General Comments
- After the references insert List of abbreviations.
- In the introduction first para very, less citations are there Need to revise it properly.
- Data Availability Statement: Not applicable.
- Check all the references carefully.
Author response: We would like to deeply thank the respected reviewer for this valuable and constructive comment. The list of abbreviations are inserted and more citations are added to paragrapgh 1 and data availability statement are added in the revised manuscript
Author action: To cope with this valuable comment, we have updated the revised manuscript as follows:
“
Revised Manuscript
”

Round 2
Reviewer 1 Report
The paper is ok.